DATA RELEASE

# Whole genome sequencing and assembly of the house sparrow, *Passer domesticus*

Vikas Kumar[1,†], Gopesh Sharma[2,3,†], Sankalp Sharma[2], Samvrutha Prasad[2], Shailesh Desai[4,†], Toral Vaishnani[5], Dalia Vishnudasan[6], Gopinathan Maheswaran[1], Kaomud Tyagi[1], Inderjeet Tyagi[1], Polavarapu B Kavi Kishor[7], Gyaneshwer Chaubey[4,*] and Prashanth Suravajhala[2,3,*]

1 Zoological Survey of India, West Bengal 700053, Kolkata, India
2 Bioclues.org, India
3 Department of Biosciences, Manipal University Jaipur, Dehmi Kalan, Rajasthan 303007, Jaipur, India
4 Department of Zoology, Banaras Hindu University, Cytogenetics Laboratory, UP 221005, Varanasi, India
5 Unipath Speciality Labs, Ahmedabad, Gujarat 380015, India
6 Amrita School of Biotechnology, Amrita Vishwa Vidyapeetham, Clappana PO 690525, Kerala, India
7 Department of Genetics, Osmania University, Hyderabad 50007, India

## ABSTRACT

The common house sparrow, *Passer domesticus*, is a small bird belonging to the family Passeridae. Here, we provide high-quality whole-genome sequencing data along with its assembly for the house sparrow. The final genome assembly was generated using a workflow that included Shovill, SPAdes, MaSuRCA, and BUSCO. The assembly consists of contigs spanning 268,193 bases and coalescing around a 922 MB sized reference genome. We used rigorous statistical thresholds to check the coverage, as the Passer genome showed considerable similarity to the *Gallus gallus* (chicken) and *Taeniopygia guttata* (Zebra finch) genomes, also providing functional annotations. This new annotated genome assembly will be a valuable resource for comparative and population genomic analyses of passerine, avian, and vertebrate evolution.

**Submitted:** 16 December 2024

\* Corresponding authors. E-mail: gyaneshwer.chaubey@bhu.ac.in; prash@bioclues.org

† Contributed equally.

Preprint submitted at https://doi.org/10.1101/2023.11.04.565608

**Subjects** Genetics and Genomics, Animal Genetics, Ecology

## INTRODUCTION

Over the past 12 years, numerous bird reference genomes have been studied, providing valuable insights into their phylogenetic relationships [1–9]. The Bird 10,000 Genomes Project provided major scientific breakthroughs in phylogenetics [10, 11]. With more than 1,200 species, comprising 13% of all known avian species, India has considerable avian diversity. However, India ranks 3rd for rare and threatened avian species worldwide [12]. The house sparrow was introduced to India via Europe from North Africa and Eurasia by the ancient Romans [13]. Sparrows are found in a variety of habitats, including grasslands, forests, deserts, agricultural areas, and urban areas, such as parks and gardens. They are omnivorous, typically feeding on insects, spiders, worms, seeds, fruits, and grains. While primarily seed eaters, they feed their young on insects and other invertebrates; hence, during breeding periods, they prefer areas rich in invertebrates [14]. Over the years, there has been a tremendous decline in their population worldwide. Understanding the mechanisms through which urbanization affects their population is limited by many

perplexing factors: rapid urbanization, deforestation, lack of cavity nesting, and absence of hedges in modern landscaping. Many hypotheses have been proposed explaining the house sparrow population decline: the increased predation by domestic cats or sparrow hawks (*Accipiter nisus*), cleaner streets reducing foraging opportunities, competition for food from other urban species, loss of nesting sites (particularly under the eaves and in the roofs of houses), pollution/air quality, in terms of both immediate and indirect toxicity through the food supply, increased use of pesticides in parks and gardens, disease transmission [15], and the Allee effect [16].

Between 1970 and 1990, common house sparrow *Passer domesticus* had a vast breeding population, with over 63 million pairs, and has been declining over the years in Turkey and certain parts of the European Union. Analyzing house sparrow habitats on a fine scale in almost 200 sites, using census data from 2003 to 2017, was one of the most complex field studies in urban Paris, with a dramatic decline of ~89% of the species over the study period [17, 18]. In India, a sharp decline in the house sparrow population was observed across Mumbai, Bengaluru, Hyderabad, and other major cities. Certain places in India experienced a decline of over 70% [19]. The lifespan of the house sparrow is 3 to 5 years in the wild and only about 20 percent of the young ones live past their first year. Cold weather and food availability decide their longevity. *Plasmodium relictum*, a parasitic infection, also affected house sparrow demography across suburban London, where sparrows have declined by 71% since 1995 [20].

Genome sequencing efforts yielded a debate over the last few years, pitting short and long read chemistries against each other. While the peacock genome yielded results [21], we earlier contemplated asking questions on the Passer genome sequencing [22]. The avian genome project [23], representing several orders, aims to resolve birds' phylogeny and collate data for testing hypotheses, understanding extinction and speciation of birds, demographic events, and their roles of drift and selection in the divergence process [24]. Recently, Magallanes-Alba *et al.* generated a pipeline for rapid genome functional-annotation anchored to the house sparrow genome re-annotation and provided transcripts [25].

Avian evolution has been of great interest in the context of extinction. Annotating birds' genomes, such as those of passerines, would enable a better understanding of their behavior and foraging traits, and further explore their evolutionary landscape.

Here, we present a genome sequencing of the house sparrow based on a muscle of the bird. We performed genome assembly and annotation using *in silico* approaches with tools that could be a valuable resource for understanding passerines' evolution, biology, ethnology, geography, and demography [26].

## METHODS

### Sample collection and genome assembly

A bird wing (muscular tissue) was taken from a male house sparrow (*P. domesticus* NCBI:txid48849) that was found dead in the lawns of the Zoological Survey of India, Kolkata, and frozen immediately in liquid nitrogen. The sample was handled by Unipath Labs for DNA extraction and sequencing using the Illumina HiSeq 2000 platform. We generated paired-end raw reads by the sequencer and used them for further downstream processing after adapter trimming. We used fastp (RRID:SCR_016962), a fast and comprehensive tool for preprocessing FASTQ files that does adapter trimming, quality filtering, read pruning,



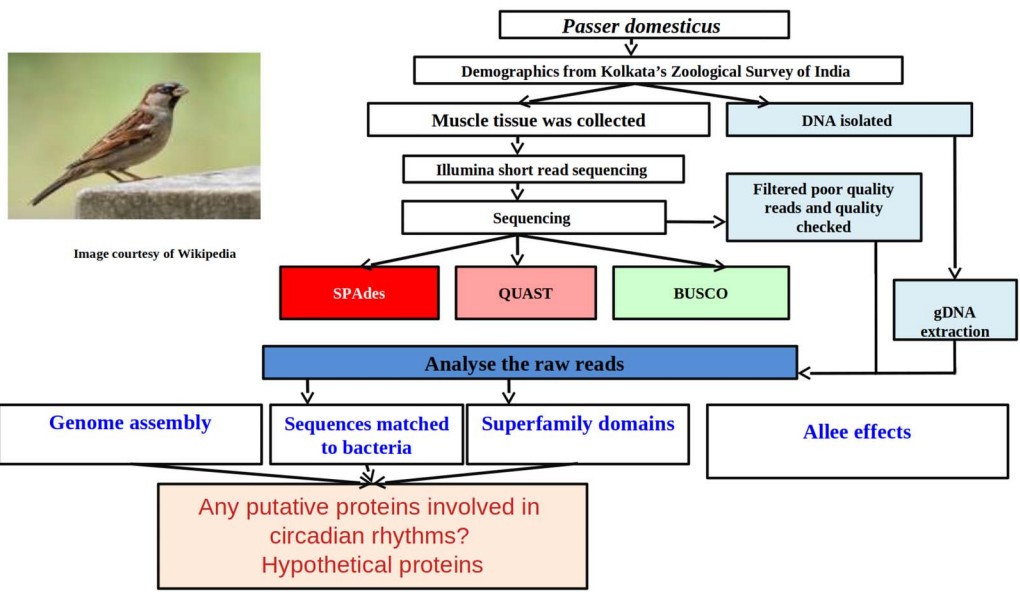

**Figure 1.** An overview of the methodology employed for annotation and assembly of *Passer domesticus*.

and the removal of duplicate reads [27]. Hence, fastp is well-suited for handling large sequencing datasets. We performed genome assembly using Shovill (version 1.1.0; RRID:SCR_017077) and SPAdes (version 3.15.4; RRID:SCR_000131) [28]. We initially mapped *Gallus gallus* (chicken) and *Taeniopygia guttata*, but the alignment was unsuccessful. Upon further alignment using bowtie2 (RRID:SCR_016368), we found 99% of the reads mapped, which led us to conduct downstream scaffolding. The gene completeness for *Passer* was assessed using BUSCO (version 5.5.0; RRID:SCR_015008) [29] and orthologous genes in the *G. gallus* genome. We performed a *de novo* assembly rather than a reference-based assembly with a specific set of nucleotide sequences used to represent an organism's genome [30]. In addition to BUSCO, we employed three tools for assembling the sample: Megahit (RRID:SCR_018551) [31], MaSuRCA (RRID:SCR_010691) [32], and SPAdes. A summary of the methods can be found in Figure 1.

## Assembly graph construction

To find read overlaps, which are crucial for contig assembly, we performed a statistical analysis of the assembly. Specifically, this passage identified the best assembly among those generated by the three assemblers (as explained above), and selected the best contigs for further downstream analysis. Statistics of the assembly were done using QUAST (RRID:SCR_001228) to identify the N50 statistics [33]. The ensuing gaps were closed using the command line tool TGS-GapCloser [34], which allowed us to close 3,150 gaps constituting the contigs present in our assembly. Prediction of genes in the genome was performed using Augustus (RRID:SCR_008417) [35], an *ab initio* hidden Markov model (HMM)-based gene prediction tool. The scaffolds of SPAdes were used as input for Augustus against trained species-specific datasets, i.e., *G. gallus* as the reference for alignment. The obtained GFF file was parsed to get the predicted coding sequence and amino acid fasta sequences.

## Mitochondrial genome assembly

We extracted the mitochondrial genome from the reads and the assembly was performed using the GetOrganelle toolkit (RRID:SCR_022963), which includes a number of scripts and libraries [36]. Next, we employed WGS read data for manipulating and disentangling assembly graphs, and generated reliable organelle genomes, accompanied by labeled assembly graphs that were visualized using Bandage (RRID:SCR_022772) [37]. The prediction of mitochondrial genes from the genome and annotation of the genome were performed using the Mitochondrial Genome Annotation Server (MITOS2) [38], which uses BLAST searches with previously annotated protein sequences to predict protein-coding genes and annotates the tRNAs apart from rRNAs present in the genome. Next, we used BLAST to identify the D-loop, a non-coding region in mitochondria that acts as the promoter for both light and heavy chains, and a key feature in mitochondria. We were able to acquire the order of the genes, their names, start and end points, and obtained the values of the intergenic regions between genes to construct the mitochondrial DNA. Mitochondrial DNA was constructed and visualized using GenomeVX [39]. To measure the codon usage and Ka/Ks, we used relative synonymous codon usage (RSCU) analysis using Molecular Evolutionary Genetics Analysis Version 11 [40].

## Phylogenetic analysis

The phylogenetic tree was constructed using MEGA11 (RRID:SCR_000667). Consensus was reached with 1Qq tree [41] and MAFFT (RRID:SCR_011811) [42] considering the following species: *Passer montanus, Passer_domesticus, Passer_ammodendri, Petronia_petronia, Pyrgilauda_blanfordi, Montifringilla_adamsi, Fringilla_polatzeki, Anthus_cervinus, Motacilla_alba, Emberiza_fucata, Spizella_passerina, Agelaius_phoeniceus, Dives_dives, Euphagus_cyanocephalus, Quiscalus_quiscula, Chrysomus_icterocephalus, Pseudoleistes_guirahuro, Molothrus_badius, Gymnomystax_mexicanus, streptopelia_orientalis_voucher_zjbj2* (outgroup).

After bootstrapping, we constructed a tree using the unweighted pair group method with arithmetic mean (UPGMA) with a gap penalty of −400 and a gap extension of 0.00. For phylogenetic tree construction, we chose Kimura distance as the substitution model with a bootstrapping set of 100, and the final tree was constructed using MAFFT. While the scoring matrix was given as BLOSUM62 AND 200 PAM, the Jukes-Cantor model was set as the substitution model and neighbor joining (NJ) for tree construction, wherein bootstrapping was set to 100. To validate the inferred tree and assess its reliability and robustness, we employed a statistical method called bootstrapping. From the bootstrapped phylogenetic tree, we concluded that a node was well supported if it remained unchanged after 95 out of 100 iterations of removing one character and resampling our tree (a bootstrap of 95% indicates this). Using the Mega X tool, we constructed a tree using the UPGMA algorithm for the Multiple Sequence Alignment (MSA) method, with a gap penalty of −400 and a gap extension of 0.00. For phylogenetic tree construction, we used the substitution parameter model Kimura-2, the maximum likelihood method, and bootstrapping values of 100. Then, a tree with MAFFT alignment was constructed, where the MSA and the tree construction were performed with a penalty score of 1.53. The sequences were aligned using a progressive method with the tree algorithm as default parameter.



## Genome annotation, comparison and statistics

Gene annotation was performed using the protein sequences, obtained from gene prediction, to annotate the genes. Repetitive regions were identified and masked prior to gene prediction using RepeatModeler (RRID:SCR_015027) [43], a *de novo* transposable element family identification and modeling package with Repeat Scout (RRID:SCR_014653) for identifying boundaries [44]. We used the Shovill contigs generated as input with 1,391 sequences (1,696,224 bp) for downstream analyses. The resulting library was later checked against *G. gallus* repeat libraries. We compared and verified the annotation using REpeat Detector (Red; version 2018.09.10), a rapid tool for detecting de novo repeats on a genomic scale [45]. As we searched for the prediction of incomplete genes at the sequence boundaries, we also aimed to predict complete genes. The Red GFF file was used as input for Augustus, which produced FASTA files containing predicted coding sequences that were used for Pfamscan searches [46]. The resulting Augustus predictions were used as input with the Pfam-A (RRID:SCR_004726) HMM library, which was manually downloaded in Stockholm format. We also queried for characteristic active site residues, if any, between the overlaps belonging to the same clans, and further checked for functional annotations and domains using InterProScan (RRID:SCR_005829) [47] to infer Gene Ontology terms. The *Acanthisitta chloris* genome was obtained from CNGB for comparing the contigs with similarity [48]. Batch Entrez, the Smith-Waterman algorithm using UVA FASTA (version 36.3.8i May, 2023) from local searches, was used for predicting proteins [49]. GffCompare was used to compare the predicted sequences at different levels of granularity, thereby annotating the sequences based on their overlaps or proximity to reference annotation transcripts [50]. Sensitivity and precision metrics were computed with the GTF file as input for generating annotated files, yielding the "super-locus" for measuring the accuracy with true positives (TP) with other features like true negatives, false negatives (FN), and false positives (FP): Sensitivity = TP/(TP+FN) and Precision = TP/(TP+FP). Finally, RefMap and TMAP files [4] were obtained measuring the reference transcript that either fully or partially matched a transcript from the GTF and those columns in the file describing the most closely matching reference transcript, respectively. We also used RagTag, a suite of tools for scaffolding and improving modern genome assemblies, to merge contigs from SPAdes [51]. FCS-GX (RRID:SCR_026367) was used to map contamination from foreign organisms using the genome cross-species aligner [52].

## Comparative divergence time estimation

In order to estimate the divergence among selected clades and species, we used BEAST V2.0 (RRID:SCR_010228) [53], where jModelTest 2 (RRID:SCR_015244) [54] was used to decide the evolutionary model based on the Bayesian Information Criterion (BIC) value, and the best suggested models were HKY+I+G. We used a mutation rate of 0.018 substitutions per site per million to estimate divergence [55]. The log files were checked using Tracer (RRID:SCR_019121), ensuring that Effective Sample Size (ESS) values were greater than 200. The final trees were then annotated with Treeannotator. Time divergences were estimated using data from all 20 taxa representing seven different families.

**Table 1.** Assembly features extracted from three different tools: Megahit, SPAdes, and MaSuRCA. According to our results, the SPAdes assembly yielded the best N50 and average value.

| Tools >>> | Megahit | SPAdes | MaSuRCA |
|---|---|---|---|
| Contigs (N) | 1,018,705 | 1,000,366 | 608,253 |
| Length | 866 MB | 922 MB | 830 MB |
| Largest contig (BP) | 83,946 | 87,277 | 78,946 |
| N50 (BP) | 1,027 | 1,343 | 740 |
| N90 (BP) | 434 | 399 | 400 |
| N's (BP) | 0.00 | 0.00 | 0.00 |
| GC % | 41.74 | 41.93 | 41.49% |

## RESULTS AND DISCUSSION

### SPAdes achieved better genome assembly statistics when compared to MaSuRCA and Megahit

We obtained the paired-end files of 14 GB each, which underwent quality check clearance. The Phred quality score (RRID:SCR_001017) was improved to 30 after Fastp (RRID:SCR_016962) trims, which was then assembled. SPAdes achieved better genome assembly statistics when compared to MaSuRCA and Megahit (Table 1). While the number of contigs was consistent between Megahit and SPAdes, MaSuRCA yielded a smaller number of contigs, as it is not referenced for bird or avian genomes. However, the GC% was found to be consistent with all three tools, attributing to an average of 771 bp. On the other hand, the N50 was comparatively better for the SPAdes assembly than the other two assemblers. In summary, the genome size of *P. domesticus* was 922 MB taking the SPAdes assembled genome into consideration.

### Gene prediction yielded 24,152 genes across as many as 45,634 transcripts

When genes were predicted using Augustus and mapped using BWA-mem from *G. gallus*, a significant similarity of 80.3% was achieved, further attributing to 24,152 genes, 38,972 introns, and 45,634 transcripts. The GFF file was then parsed to check downstream functional annotations using the KOG, non-redundant (NR), and Uniprot databases. The consensus hits were searched with "uniq sort" to find the most occurring species, and we selected the 10 best hits. We observed that NR yielded the best results with *Passer montanus* with 7,308 occurencies, *Stutzerimonas stutzeri* with 1,365, *Melospiza melodia maxima* with 939, *Pyrgilauda ruficollis* with 788, *Hirundo rustica rustica* with 780, *Chloebia gouldiae* with 715, *Limosa lapponica baueri* with 679, *Onychostruthus taczazanowskii* with 652, *Lonchura striata domestica* with 457, and *Motacilla alba* with 319. Surprisingly, we also detected bacterial sequences among the hits [56, 57]. We also identified many hypothetical proteins, corresponding to previously unknown regions, which supports the need to annotate the genome more comprehensively. We identified numerous genes crucial for adaptations, including those involved in vocal learning, circadian rhythms, and Allee effects. For instance, BMAL1 clock genes, known to regulate circadian rhythms, were identified in our list. We also found TLR4, which helps identify pathogens and initiate immune responses, in addition to HBAA, HBB, and HBA, which are known for oxygen transport, thereby also supporting adaptation.

Avian genomes are about 70% smaller than mammalian genomes, and Passerines, being small, are thought to maintain the contiguity that we have shown as compared to other



vertebrates. We are limited by the number of genomic attributes coming from non-coding regions; nevertheless, this could be due to the large number of hypothetical proteins that require accurate annotation. While many genomes show synteny, we are planning to sequence more genomes that could provide insights into adaptation, evolutionary mechanisms, substitutions, and, importantly, circadian rhythms and the electromagnetic effects that these birds might be affected by. The latter two may have contributed significantly to their drifting away from urban life.

The 710 contigs were mapped to GenBank, yielding 152 hypothetical proteins. This result indicates that a large number of proteins emerged from the known-unknown regions, implying the need to annotate the genome in more detail [58]. Among these, 252 sequences from protein domains matched to animals while the rest matched to bacteria. We observed a large number of largely conserved Kelch domains in addition to a PHD finger 1 in Histone-lysine N-methyltransferase 2C (KMT2C) and 2D (KMT2D) protein. The myeloid/lymphoid or mixed-lineage leukemia protein 3 (MLL3) is associated with circadian factors contributing to genome-scale circadian transcription. While these observations could indicate that a large number of genes have undergone divergent evolution or that some genes were lost and subsequently regained through speciation, the fragmentation of this genome aligns with BUSCO statistics reported earlier for avian genomes, including those for the Chaffinch [59].

Our sequence annotation is limited but diverse, as evidenced by higher gene counts compared to *G. gallus* [60]. However, many genes in the Chicken genome remain unmapped and unidentified. Ours is a draft genome; nonetheless, for small species such as Passerines functional diversity is poorly understood, and evidence can only be obtained through experimentation. While we attempted to create a high-quality *de novo*-based reference genome, our final assembly encompassing 922 MB is in agreement with the Elgvin *et al.* [61] works of a similarly sized genome, albeit the medium-density linkage map and order. The assembly of scaffolds into chromosomes was precisely done across several samples; however, we were unable to achieve this due to limited sample acquisition. Nevertheless, the genes were mapped to their reference genome for downstream analyses (Table 2). Our BUSCO result indicates a comparatively good assembly, with 747 complete BUSCOs (C), comprising 744 complete and single-copy BUSCOs (S). Although we found 3 complete and duplicated (D) and 603 fragmented BUSCOs (F), the distribution of BUSCO categories suggests that the assembly is complete [62]. Furthermore, RagTag yielded desirable contiguity of sequences in the form of scaffolds, with 623,567 placed sequences totaling 362,669,528 bp, 23,433 unplaced sequences comprising 16,110,716 bp, and an overall 623,407 bp of gap sequences.

## Seven superfamily domains are conserved between *P. domesticus* and *Acanthisitta chloris*

The final set of contigs obtained for assembly was mapped against NCBI taxa [45]. This mapping revealed matches for 14,947 contigs, with 769 of them achieving 100% query coverage across various taxa. Amongst the superfamily members, the intermediate filament protein Brain Acid Soluble Protein 1 (BASP1), glycine-rich LPXTG-anchored collagen-like adhesin, collagen with keratin, and some hypothetical protein-FTsK, translocase domains. What remains intriguing is the presence of a probable chromatin-remodeling complex ATPase chain, which is possibly associated with Allee

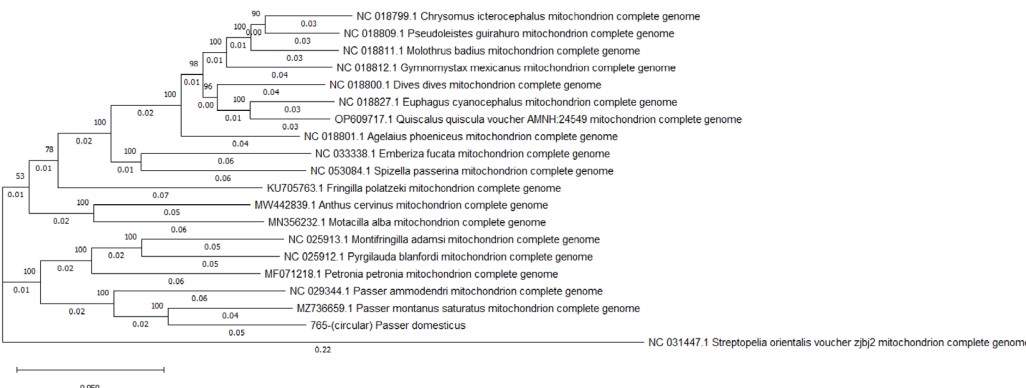

**Figure 2.** Phylogenetic tree representing all 20 species to which *P. domesticus* is closely related, including *P. montanus*, *P. ammodendri*, *Petronia petronia*, *Pyrigilauda blandfordi*, and *Montifringilla adamsi*. All these species have highly supported bootstrap values (i.e., 100). Whereas less closely related species, such as *Motacilla alba* and *Anthus cervinus*, all belong to branches with a low bootstrap value (i.e., 78). *streptopelia_orientalis_voucher_zjbj2* shows the most distant species in the tree, which are the outgroup species with a low supported bootstrap value of 53.

**Table 2.** Comparison of genes that were mapped to the reference genome.

| Sl. No. | Gene name | NCBI accession | Contigs mapped |
|---------|-----------|----------------|----------------|
| 1 | APC | NC_087512.1:52445657-52540315 | NODE_14189_length_3730_cov_2.598686 |
| 2 | HSDL2 | NC_087512.1:69765571-69788616 | NODE_20005_length_3394_cov_4.411215 |
| 3 | ETFA | NC_087487.1:960048-989096 | NODE_45445_length_2642_cov_2.323977 |
| 4 | NOTCH2 | NC_087480.1:32766603-32849060 | NODE_2951_length_5290_cov_3.101669 |
| 5 | NOTCH1 | NC_087491.1:10550409-10592339 | NODE_82034_length_2127_cov_3.214634 |
| 6 | GLI3 | NC_087474.1:51942133-52148229 | NODE_5580_length_4619_cov_3.460810 |
| 7 | WNT4 | NC_087495.1:277607-294952 | NODE_126693_length_1756_cov_2.892198 |
| 8 | TIMP2 | NC_087493.1:1552785-1561149 | NODE_92905_length_2020_cov_2.762738 |
| 9 | RBP1 | NC_087484.1:25002122-25019513 | NODE_19406_length_3423_cov_2.725344 |
| 10 | PRK11A | NC_087512.1:13609738-13635725 | NODE_48977_length_2574_cov_3.450941 |
| 11 | GNAI1 | NC_087478.1:9772700-9805360 | NODE_5061_length_4724_cov_3.114698 |
| 12 | CRHR1 | NC_087500.1:437197-467780 | NODE_32621_length_2944_cov_3.164632 |
| 13 | ZFAND5 | NC_087512.1:62564205-62578406 | NODE_86240_length_2083_cov_4.130608 |
| 14 | SECISBP2 | NC_087512.1:42294093-42319272 | NODE_2582_length_5428_cov_3.587927 |
| 15 | REEP5 | NC_087512.1:52407038-52426127 | NODE_48707_length_2579_cov_3.693046 |
| 16 | MIA3 | NC_087476.1:85960932-85989020 | NODE_12474_length_3848_cov_3.843543 |
| 17 | A2M | NC_087475.1:87627899-87657608 | NODE_583_length_7451_cov_3.871576 |
| 18 | KCNN2 | NC_087512.1:68344752-68419555 | NODE_16813_length_3560_cov_3.043928 |
| 19 | BNC2 | NC_087512.1:33461143-33796687 | NODE_5333_length_4668_cov_3.558048 |

effects that occur when individual fitness suffers in populations that are small or sparse [16]. We also suggest that the genetic architecture and community-wide admixture could provide insights into their evolution [58].

On divergence, Columbidae appeared to have diverged early compared to others and were considered an outgroup in this analysis. In our sequenced sample, the house sparrow belonged to the family Passeridae. The divergence and expansion of major families began around 7–9 million years ago. Our studied sample formed a subclade with the Eurasian tree sparrow (*P. montanus saturatus*) and the Saxual sparrow (*P. ammodendri*), having a Time to Most Recent Common Ancestor (TMRCA) of 4.4 million years ago. Furthermore, our analysis yielded a TMRCA of 2.9 million years ago for *P. domesticus* and *P. m. saturatus* (Figure 2).

**Table 3.** All the genes, except atp6, with ka/ks values below one, indicating they are undergoing strong purifying selection. Conversely, atp6 undergoes a positive selection with a ka/ks value greater than one.

| Genes | Ka/ks |
|-------|-------|
| cox2 | 0.81 |
| atp6 | 1.03 |
| atp8 | 0.429 |
| cox3 | 0.012 |
| nad3 | 0.02 |
| nad4l | 0.035 |
| nad5 | 0.03 |
| nad4 | 0.13 |
| cob | 0.02 |
| nad6 | 0.048 |
| nad1 | 0.034 |
| nad2 | 0.017 |
| cox1 | 0.003 |

## The mitochondrial genome yielded a genome of size 16,804 bp

Gene annotation predicted 37 genes, including 13 protein-coding genes, 22 tRNAs, 2 rRNAs, and the D-loop non-coding region. A labeled mitogenome was produced using the GenomeVX online platform (Figure 2). The RSCU analysis of the codons was calculated, and the lowest preferences were AGA (R), AGG (R), and ACG (T) (Table 2). Although these codons are less frequently used due to lower levels of their specific tRNAs, they may slow down protein translation efficiently, thus avoiding the use of these codons in efficient protein translation. We observed different patterns of selective pressure by analyzing the Ka/Ks ratios for the specified genes with strong purifying selection seen in the majority of genes, which have Ka/Ks ratios much less than 1, including cox1 (0.003), Cox3 (0.012), Nad2 (0.017), Nad3 (0.02), cob (0.02), NAD1 (0.034), Nad4l (0.035), Nad5 (0.03), and Nad6 (0.048) (Table 3). Given that the proteins encoded by these genes are essential to the organism's survival and functionality, and that harmful mutations are progressively eliminated, these genes are likely highly conserved. The Atp6 gene, on the other hand, appears to be under positive selection, indicating that natural selection may favor modifications in the protein derived from this gene that may be advantageous, with a Ka/Ks ratio of 1.03 (Table 4; Figure 3). While the scoring matrix was specified as BLOSUM62 AND 200 PAM, the Jukes-Cantor model was set as the substitution model and NJ for tree construction, with bootstrapping set to 100 (Figure 4).

## CONCLUSIONS

We provide here a whole genome sequencing and assembly for the house sparrow, *P. domesticus*, which could help answer many genomic questions. Our annotation serves as a valuable resource for checking adaptation, divergence, and speciation. The orthologous annotation and protein mapping, along with the bacterial correlates, suggest that these are crucial for organismal survival and function.

## DATA AVAILABILITY

The genome assembly has been deposited at NCBI under BioProject ID: PRJNA1027087 and SRR26357069. Additional data is available in the GigaDB repository [58].

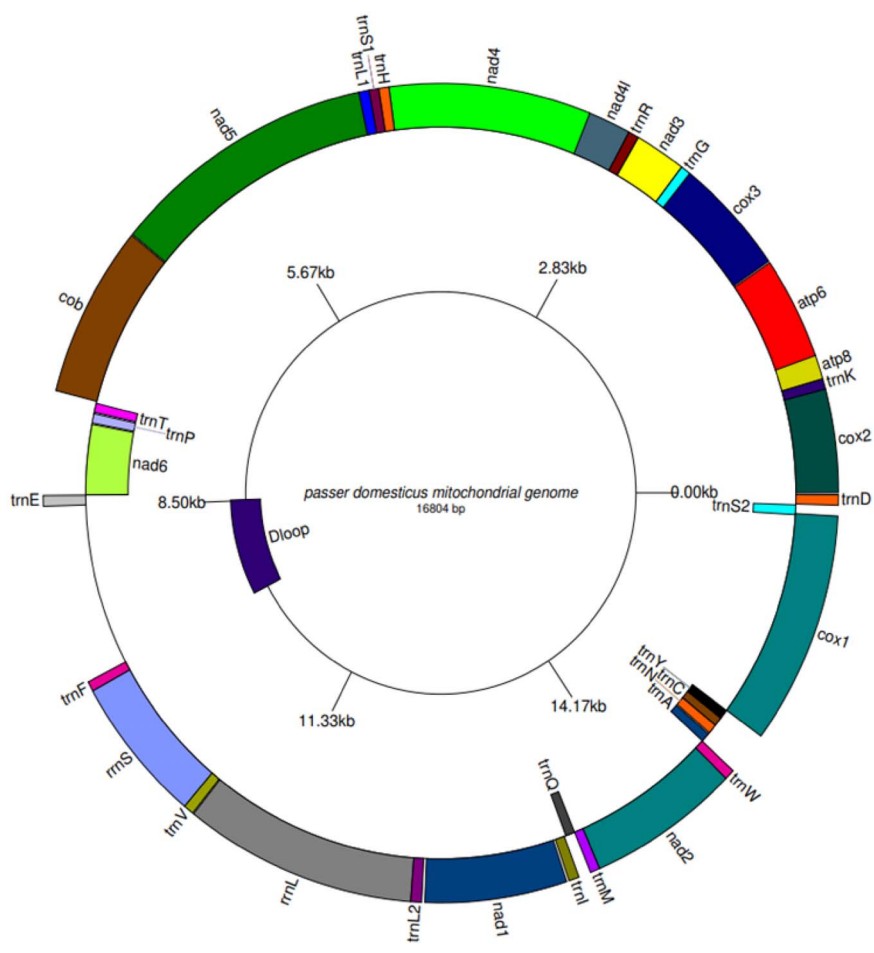

**Figure 3.** Labeled mitogenome representing 37 genes, including the protein-coding genes tRNAs, rRNAs, and the non-coding region, which is the control region or the d-loop.

## LIST OF ABBREVIATIONS

FN: false negatives; FP: false positives; HMM: hidden Markov model; MSA: Multiple Sequence Alignment; NJ: neighbor joining; NR: non-redundant; Red: REpeat Detector; RSCU: relative synonymous codon usage; TN: true negatives; TP: true positives; TMRCA: Time to Most Recent Common Ancestor; UPGMA: unweighted pair group method with arithmetic mean.

## DECLARATIONS

### Ethics approval and consent to participate

The authors declare that ethical approval was not required for this type of research.

### Competing interests

The authors declare that they have no competing interests.

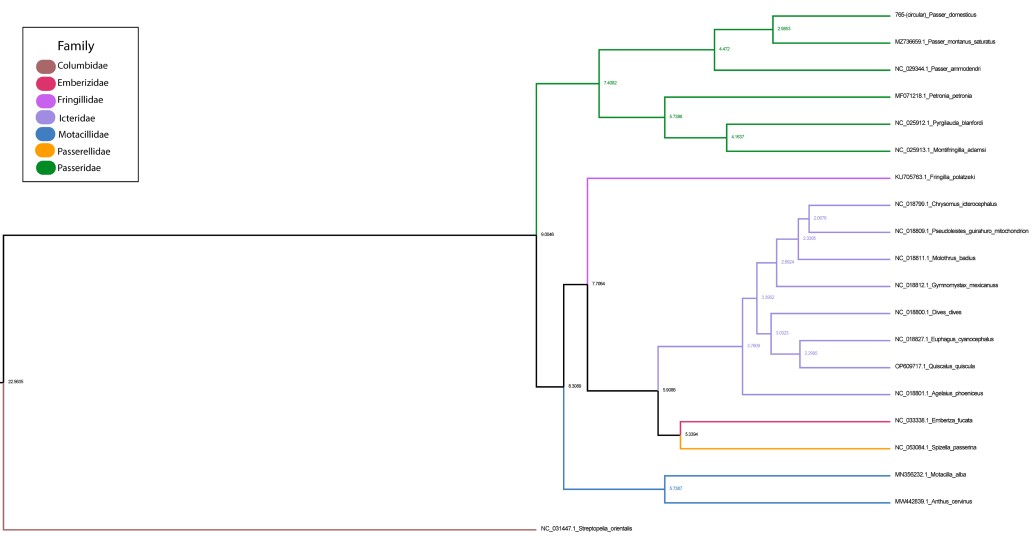

**Figure 4.** A time-calibrated tree with 20 selected taxa using complete mitochondrial DNA sequences. The tree was generated using Beast v2.5 with 20 million MCMC runs. Each respective family is represented by a different color, and the node labels indicate the age in million years.

**Table 4.** The RSCU values provided for the mitochondrial genome, with the codons CUA (L), GCC(A), and CGA(R) having the highest values, indicate a more frequent occurrence. They may be preferred due to the abundance of tRNA, which contributes to efficient protein translation and protein stability, and reflects evolutionary adaptation for protein synthesis. The codons that are translated efficiently are preferred in highly expressed genes.

| Codon | RSCU | Codons | RSCU | Codons | RSCU | Codons | RSCU |
|---|---|---|---|---|---|---|---|
| UUU(F) | 0.27 | UCU(S) | 0.53 | UAU(Y) | 0.44 | UGU(C) | 0.26 |
| UUC(F) | 1.73 | UCC(S) | 2.08 | UAC(Y) | 1.56 | UGC(C) | 1.74 |
| UUA(L) | 0.47 | UCA(S) | 1.83 | UAA(*) | 0.28 | UGA(*) | 2.69 |
| UUG(L) | 0.18 | UCG(S) | 0.35 | UAG(*) | 0.03 | UGG(W) | 1 |
| CUU(L) | 0.34 | CCU(P) | 0.45 | CAU(H) | 0.32 | CGU(R) | 0.49 |
| CUC(L) | 1.23 | CCC(P) | 1.42 | CAC(H) | 1.68 | CGC(R) | 1.15 |
| CUA(L) | 3.08 | CCA(P) | 1.84 | CAA(Q) | 1.64 | CGA(R) | 3.7 |
| CUG(L) | 0.71 | CCG(P) | 0.29 | CAG(Q) | 0.36 | CGG(R) | 0.49 |
| AUU(I) | 0.52 | ACU(T) | 0.58 | AAU(N) | 0.31 | AGU(S) | 0.14 |
| AUC(I) | 1.61 | ACC(T) | 1.71 | AAC(N) | 1.69 | AGC(S) | 1.07 |
| AUA(I) | 0.86 | ACA(T) | 1.67 | AAA(K) | 1.7 | AGA(R) | 0.08 |
| AUG(M) | 1 | ACG(T) | 0.04 | AAG(K) | 0.3 | AGG(R) | 0.08 |
| GUU(V) | 0.83 | GCU(A) | 0.48 | GAU(D) | 0.29 | GGU(G) | 0.38 |
| GUC(V) | 0.93 | GCC(A) | 2.22 | GAC(D) | 1.71 | GGC(G) | 0.95 |
| GUA(V) | 1.64 | GCA(A) | 1.19 | GAA(E) | 1.47 | GGA(G) | 1.9 |
| GUG(V) | 0.61 | GCG(A) | 0.11 | GAG(E) | 0.53 | GGG(G) | 0.77 |

## Authors' contributions

PS ideated the project with GC and PBK. VK, SS, SP, AT, GS, DV, SD, and PS contributed equally to the work. All the other authors worked to improve the tables and figures. PS, GC, and PBK proofread the manuscript.

## Funding

No external funding was used in this work.

## Acknowledgements

PS gratefully acknowledges the Department of Forestry, Governments of Rajasthan and India, Zoological Survey of India, and Birla Institute of Scientific Research for providing the a priori permission for the collection of the bird specimen. PS, GS and SS gratefully acknowledge the support of veterinarians V Galav and S Srivastava for help in collection of samples.

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
