## [Editor Report]

Editor’s AssessmentThis paper presents present the genome sequencing of the house sparrow (Passer domesticus) carrying out genome assembly and annotation using in silico approaches with tools that could be a valuable resource for understanding passerine evolution, biology, ethnology, geography, and demography. The final genome assembly was generated using short read sequencing and a computational workflow that included Shovill, SPAdes, MaSuRCA, and BUSCO benchmarking. Producing a 922 MB reference genome with 24,152 genes. The first draft was significantly smaller than this but peer review provided suggestions on how to improve the assembly quality. And after a few attempts and assembly with a reasonable size and BUSCO score was achieved. This openly available data potentially serving as a valuable resource for checking adaptation, divergence, and speciation of birds.Editor’s AssessmentThis paper presents present the genome sequencing of the house sparrow (Passer domesticus) carrying out genome assembly and annotation using in silico approaches with tools that could be a valuable resource for understanding passerine evolution, biology, ethnology, geography, and demography. The final genome assembly was generated using short read sequencing and a computational workflow that included Shovill, SPAdes, MaSuRCA, and BUSCO benchmarking. Producing a 922 MB reference genome with 24,152 genes. The first draft was significantly smaller than this but peer review provided suggestions on how to improve the assembly quality. And after a few attempts and assembly with a reasonable size and BUSCO score was achieved. This openly available data potentially serving as a valuable resource for checking adaptation, divergence, and speciation of birds.

---

## [Reviewer Report]

Indicate in the comments box below whether you are happy with the changes made or if the manuscript is unacceptable.Comments on revised manuscriptThe authors have subtly improved the original version previously presented, but have not managed to surpass the minimum standards established by the publisher to be published by the journal. Easily achievable changes have been requested to complement the analysis previously made and have been ignored. Requests have not been answered, graphics that generate confusion between them and the text presented have not been fixed, and no relevant improvement between the previous and current versions has been shown.

---

## [Reviewer Report]

Indicate in the comments box below whether you are happy with the changes made or if the manuscript is unacceptable.Comments on revised manuscriptThe authors used FCS-GX to exclude contaminating sequences in the genome, so I agree that this paper should be published.

---

## [Reviewer Report]

Reviewer name and names of any other individual's who aided in reviewer Agustin Ariel BaricallaDo you understand and agree to our policy of having open and named reviews, and having your review included with the published papers. (If no, please inform the editor that you cannot review this manuscript.)YesIs the language of sufficient quality?YesPlease add additional comments on language quality to clarify if needed
Are all data available and do they match the descriptions in the paper? NoAdditional CommentsMatching data: NCBI project with access to the NCBI-SRA deposited raw data. Nonmatching data: Oxford Nanopore data: The authors reply to a previously submitted manuscript arguing that this data was not used, but Fig. 1 refers to Nanopore Minion data. The manuscript body and the additional data section do not include the Quast and BUSCO reports or their corresponding plots.Are the data and metadata consistent with relevant minimum information or reporting standards? See GigaDB checklists for examples <a href="http://gigadb.org/site/guide" target="_blank">http://gigadb.org/site/guide</a>NoAdditional CommentsGigaByte suggests a checklist including the genome, CDS, and proteins in FASTA format, as well as the annotations in GFF format; however, these items are not available for evaluation.Is the data acquisition clear, complete and methodologically sound?YesAdditional CommentsIs there sufficient detail in the methods and data-processing steps to allow reproduction?YesAdditional CommentsThe FastP step for raw data processing is mentioned in the results section but is not detailed in the methods section.Is there sufficient data validation and statistical analyses of data quality? NoAdditional CommentsThe authors have not included the BUSCO results. The OrthoDB database for 'passeriformes_odb12' contains over 10,000 curated genes, representing approximately 50-60% of the total genes in a typical passeriform genome. Therefore, the BUSCO report for the new assembly should be provided. The author mentioned that "The gene completeness for Passer was assessed through Benchmarking Universal Single-Copy Orthologs ( Busco version 5.5.0 ) [26] by using the orthologous genes in the Gallus gallus [ chicken] genome" but BUSCO uses the OrthoDB datasets to run, I do not understand what this phrase refers to.Is the validation suitable for this type of data?YesAdditional CommentsIs there sufficient information for others to reuse this dataset or integrate it with other data?YesAdditional CommentsAll the procedures are consistent and the programs or pipelines are well-known and well documented in the bioinformatic and genomic fields.Any Additional Overall Comments to the AuthorThe inclusion of the mitochondrial genome represents a significant improvement in this manuscript. I recommend presenting all nuclear results together first, followed by a separate and clear description of the mitochondrial analysis and findings to enhance clarity. The data is interesting for analyzing the genetic dynamics behind Passer domesticus adaptation and evolution and can show differences between the previous genomes available from a European reference sample but this is not presented in this work. As of this revision, the NCBI's Passer domesticus genome includes two European reference genomes, both classified with 'chromosome-like' status (NCBI: GCF_036417665.1 and GCA_001700915.1). These genomes can be utilized in two distinct ways: (1) performing a 'genome-guided assembly' with MASURCA, using one of these genomes alongside the Illumina data, or (2) conducting genome scaffolding by employing one of these genomes as a reference and the assembled genome from raw reads as a query, using tools like RagTag or the chromosome scaffolder available in MASURCA. Both approaches could potentially lead to improvements in scaffold number and contiguity metrics, such as N50, N90, and the largest scaffold.RecommendationMajor Revision

---

## [Reviewer Report]

Reviewer name and names of any other individual's who aided in reviewer Gang WangDo you understand and agree to our policy of having open and named reviews, and having your review included with the published papers. (If no, please inform the editor that you cannot review this manuscript.)YesIs the language of sufficient quality?YesPlease add additional comments on language quality to clarify if needed
There are many details in the article, such as citation format, spelling, etc. [Supplementary Table 3a, 3b, 3c) → (Supplementary Table 3a, 3b, 3c) The citation format of the article also needs to be adjusted according to the journal requirements.Are all data available and do they match the descriptions in the paper? YesAdditional CommentsAre the data and metadata consistent with relevant minimum information or reporting standards? See GigaDB checklists for examples <a href="http://gigadb.org/site/guide" target="_blank">http://gigadb.org/site/guide</a>YesAdditional CommentsIs the data acquisition clear, complete and methodologically sound?YesAdditional CommentsIs there sufficient detail in the methods and data-processing steps to allow reproduction?NoAdditional CommentsA previous reviewer mentioned that RagTag could be used to improve the quality of genome assembly. I suggest you seriously consider this.Is there sufficient data validation and statistical analyses of data quality? YesAdditional CommentsIs the validation suitable for this type of data?YesAdditional CommentsIs there sufficient information for others to reuse this dataset or integrate it with other data?NoAdditional CommentsAny Additional Overall Comments to the AuthorThe article is logically clear and the analysis is complete. The description of both sample collection and sequencing is relatively clear. At the same time, the analysis process shown in Figure 1 is also very reasonable. However, as described by the previous reviewer, I suggest that you remove the high-quality level. There are many details in the article, such as citation format, spelling, etc. [Supplementary Table 3a, 3b, 3c) → (Supplementary Table 3a, 3b, 3c) The citation format of the article also needs to be adjusted according to the journal requirements. Figure 2, the letters of a and b are too different, please unify them. Figure 4 is completely unclear, please increase the font size. A previous reviewer mentioned that RagTag could be used to improve the quality of genome assembly. I suggest you seriously consider this.RecommendationMajor Revision